# Discovering alternative solutions beyond the simplicity bias in recurrent neural networks

**William Qian**[1,2], **Cengiz Pehlevan**[2,3,4]
[1]Biophysics Graduate Program
[2]Kempner Institute for the Study of Natural and Artificial Intelligence
[3]John A. Paulson School of Engineering and Applied Sciences
[4]Center for Brain Science
Harvard University, Cambridge MA 02138
`williamqian@g.harvard.edu, cpehlevan@seas.harvard.edu`

## Abstract

Training recurrent neural networks (RNNs) to perform neuroscience-style tasks has become a popular way to generate hypotheses for how neural circuits in the brain might perform computations. Recent work has demonstrated that task-trained RNNs possess a strong simplicity bias. In particular, this inductive bias often causes RNNs trained on the same task to collapse on effectively the same solution, typically comprised of fixed-point attractors or other low-dimensional dynamical motifs. While such solutions are readily interpretable, this collapse proves counterproductive for the sake of generating a set of genuinely unique hypotheses for how neural computations might be performed. Here we propose Iterative Neural Similarity Deflation (INSD), a simple method to break this inductive bias. By penalizing linear predictivity of neural activity produced by standard task-trained RNNs, we find an alternative class of solutions to classic neuroscience-style RNN tasks. These solutions appear distinct across a battery of analysis techniques, including representational similarity metrics, dynamical systems analysis, and the linear decodability of task-relevant variables. Moreover, these alternative solutions can sometimes achieve superior performance in difficult or out-of-distribution task regimes. Our findings underscore the importance of moving beyond the simplicity bias to uncover richer and more varied models of neural computation.

## 1 Introduction

Developing recurrent models of neural computations has become an increasingly popular approach to generate hypotheses for neuroscience (Mante et al., 2013; Rajan et al., 2016; Maheswaranathan et al., 2019; Yang et al., 2019; Sylwestrak et al., 2022; Daie et al., 2023; Beiran et al., 2023; Nair et al., 2023; Driscoll et al., 2024; Javadzadeh et al., 2024; Genkin et al., 2025). In particular, recurrent neural networks (RNNs) trained on neuroscience-style tasks offer insight into possible solutions that may be implemented at an approximate level by biological neural circuits. Such RNNs are typically trained via backpropagation through time (Werbos, 1990) or FORCE (Sussillo & Abbott, 2009), methods that seem to bear little resemblance to the way learning proceeds in biological circuits (Crick, 1989; Lillicrap et al., 2020). Nonetheless, resemblances between solutions found by artificial and biological networks have the potential to shed light on shared principles of neural computation that emerge despite these differences (Mante et al., 2013; Yamins et al., 2014; Sussillo et al., 2015; Kell et al., 2018; Banino et al., 2018; Schrimpf et al., 2020; Feather et al., 2023; Jensen et al., 2024; Pagan et al., 2025).

Central to this research program is the ability to produce multiple competing hypotheses that can then be evaluated on equal footing via comparisons against experimental data (Barak et al., 2013; Sussillo et al., 2015; Soldado-Magraner et al., 2024; Pagan et al., 2025; Huang et al., 2025). Ideally, training multiple RNNs on a particular task would be sufficient to yield a diverse range of solutions for this purpose. Yet, this strategy faces major obstacles in scenarios where training procedures overwhelmingly bias RNNs towards particular kinds of solutions.

Recent work has shown that task-trained RNNs exhibit a bias towards simple solutions—solutions that use a minimal arrangement of low-dimensional dynamical structures such as fixed point attractors and limit cycles, and reuse dynamical motifs where possible (Turner & Barak, 2023; Driscoll et al., 2024; Hazelden et al., 2025). These types of solutions have desirable properties including parsimony and flexibility, and often lend themselves to relatively straightforward interpretation via analysis techniques such as targeted dimensionality reduction, low-rank approximation, and dynamical systems analysis (Sussillo & Barak, 2013; Mante et al., 2013; Dubreuil et al., 2022; Valente et al., 2022; Driscoll et al., 2024). However, for many neuroscience-style tasks, this simplicity bias can be strong enough to cause different networks trained on the same task to collapse to effectively the same, minimal solution, a phenomenon referred to as dynamic collapse (Hazelden et al., 2025). Despite the desirable properties of such solutions, it remains far from clear that this bias towards simplicity is always aligned with the inductive biases of biological circuits. For example, RNNs trained on simple memory tasks ubiquitously find solutions using persistent activity held in stable attractor states (Maheswaranathan et al., 2019; Turner & Barak, 2023; Driscoll et al., 2024; Hazelden et al., 2025), yet population-level recordings have shown that the neural representations underlying memory functions can be highly dynamic (Spaak et al., 2017; Lundqvist et al., 2018; Daie et al., 2023; Ritter & Chadwick, 2025). These observations raise an important question: how can RNNs be trained to generate unique hypotheses for recurrent computations that go beyond the simplicity bias?

The most natural toolkit for generating different task solutions includes varying hyperparameters such as the initialization scale, training seed, and model architecture. The initialization scale in particular has been shown to affect lazy versus rich learning in RNNs (Schuessler et al., 2020; Liu et al., 2023; Bordelon et al., 2025), as well as the emergence of "aligned" or "oblique" solutions (Schuessler et al., 2024). However, dynamic collapse can still be observed even when RNNs are initialized in the highly chaotic regime (Hazelden et al., 2025). While varying these basic knobs is sometimes sufficient to generate a multitude of qualitatively distinct solutions, (Turner et al., 2021; Huang et al., 2025; Murray, 2025; Kurtkaya et al., 2025), many classes of realistic solutions are likely still inaccessible through these means. For instance, Pagan et al. (2025) found that a large population of RNNs trained on the same context-dependent decision making task populated only one corner of the solution space compatible with neural data. Moreover, solutions obtained by varying architectural details can appear representationally distinct, but often implement the same underlying dynamical solution, as revealed by fixed-point topology (Maheswaranathan et al., 2019).

In this paper, we propose a simple method for generating unique solutions to RNN tasks, extending beyond solutions discoverable by standard means. This method, which we call Iterative Neural Similarity Deflation (INSD), is loosely analogous to the Gram-Schmidt procedure but in the space of RNN solutions. By iteratively penalizing the linear predictivity of neural activity produced by previously trained RNNs in an online fashion, we find solutions that diverge from the prototypical solutions to classic neuroscience-style tasks. We show that the alternative solutions generated in this manner not only use distinct representational geometry as expected, but also use different dynamical motifs and encode task variables more nonlinearly. Across all tasks, these solutions forgo the usage of fixed point attractors and slow manifolds for keeping track of task-relevant information, and instead tend to maintain task-relevant information in dynamically evolving subspaces of activity. Surprisingly, we find that these alternative solutions can sometimes achieve superior performance when tested in difficult out-of-distribution task conditions.

## 2 METHODS

### 2.1 SETUP AND TRAINING PROCEDURES

We consider rate-based RNNs obeying the dynamics

$$\frac{d\boldsymbol{x}}{dt} = -\boldsymbol{x} + \boldsymbol{W}\boldsymbol{r} + \boldsymbol{J}^{\text{in}}\boldsymbol{u}(t) \tag{1}$$

where $\boldsymbol{x} \in \mathbb{R}^N$ represent neural activations over $N$ units, $\boldsymbol{W} \in \mathbb{R}^{N \times N}$ is the recurrent weight matrix, $\boldsymbol{J}^{\text{in}} \in \mathbb{R}^{N \times N_{in}}$ and $\boldsymbol{u}(t) \in \mathbb{R}^{N_{in}}$ are the input weights and inputs, respectively, $\boldsymbol{r} = \phi(\boldsymbol{x})$

are the "firing rates", and $\phi$ is an elementwise nonlinearity which we take to be $\tanh$. The output is given by $\boldsymbol{y}(t) = \boldsymbol{J}^{\text{out}}\boldsymbol{r}(t)$, for readout weights $\boldsymbol{J}^{\text{out}} \in \mathbb{R}^{N_{out} \times N}$.

For each task, we first train a reference RNN to minimize the mean squared error

$$\mathcal{L} = \frac{1}{T} \int_0^T \|\boldsymbol{y}(t) - \boldsymbol{y}^\star(t)\|^2 dt, \tag{2}$$

averaged over different input conditions $\boldsymbol{u}(t)$, via batch gradient descent over the parameters $\Theta = \{\boldsymbol{W}, \boldsymbol{J}^{\text{in}}, \boldsymbol{J}^{\text{out}}\}$. We initialize the recurrent weights as $\boldsymbol{W}_{ij} \sim \mathcal{N}(0, g^2/N)$, where $g$ is a gain parameter. The input and output weights are both initialized with entries drawn from $\mathcal{N}(0, 1/N)$.

We then apply a neural activity similarity penalty to subsequent RNNs trained on the same task. In particular, for each batch of input conditions, firing rates $\boldsymbol{R}_1 \in \mathbb{R}^{(BL_t) \times N}$ and $\boldsymbol{R}_2 \in \mathbb{R}^{(BL_t) \times N}$ are collected from the reference RNN and the second RNN, respectively, where the batch ($B$) and discrete timestep ($L_t$) dimensions have been flattened. These firing rates are then projected into their respective readout null spaces, yielding $\boldsymbol{R}_1^\perp$ and $\boldsymbol{R}_2^\perp$. The second RNN is then trained with the loss

$$\mathcal{L}' = \mathcal{L} + \lambda S(\boldsymbol{R}_2^\perp, \boldsymbol{R}_1^\perp), \tag{3}$$

where $S$ is some neural similarity measure, and $\lambda$ is a hyperparameter representing the strength of the penalty. We project firing rates to readout nullspaces prior to applying the similarity penalty because allowing it to operate on the output potent component of activity would be counterproductive to solving the task. In particular, if the reference RNN achieves near perfect outputs $\boldsymbol{y}(t) \approx \boldsymbol{y}^\star(t)$, then to achieve similar task performance, the second RNN's activity must necessarily be able to linearly predict the output potent component of the reference RNN's activity. This procedure can be continued iteratively, with a third RNN penalized with respect to both previous RNNs via a loss

$$\mathcal{L}'' = \mathcal{L} + \lambda \left[ S(\boldsymbol{R}_3^\perp, \boldsymbol{R}_1^\perp) + S(\boldsymbol{R}_3^\perp, \boldsymbol{R}_2^\perp) \right], \tag{4}$$

and so on. We refer to this procedure as Iterative Neural Similarity Deflation (INSD), and label RNNs trained in this manner alt-1, alt-2, etc. This approach for explicitly encouraging different task solutions somewhat resembles the Barlow Twins method (Zbontar et al., 2021) in computer vision and the method of linear adversarial concept erasure (Ravfogel et al., 2022) in algorithmic fairness.

For comparison, we also train a population of "standard" RNNs on each task. For simplicity, we use the same architecture for all RNNs, training ten RNNs with different seeds for each initialization scale $g \in [0.01, 0.5, 1.0, 1.5]$. A more detailed sweep including architecture, hyperparameters, and nonlinearities can be found in Maheswaranathan et al. (2019). Training details are specified in A.1.

## 2.2 NEURAL SIMILARITY MEASURES

There exists a large variety of neural similarity measures that could be used for the similarity penalty, each with their own advantages and drawbacks (Raghu et al., 2017; Kornblith et al., 2019; Williams et al., 2021; Harvey et al., 2024a; Williams, 2024; Cloos et al., 2024; Harvey et al., 2024b). For our purposes, we seek a metric which is invariant to relabeling or rotation of neural axes, and for which forwards and backwards passes can be efficiently computed online.

For many neural similarity measures, solving a task while maintaining low neural similarity with respect to a reference network admits a trivial yet undesirable solution: a subspace of activity implements a version of the reference solution, while the remaining degrees of freedom simply inflate the dimensionality of the neural activity with task-irrelevant dynamics. In particular, centered kernel alignment, representational similarity analysis (RSA), and linear predictivity scores in the direction of [reference RNN → penalized RNN] can all be driven arbitrarily close to $0$ in this manner (see A.2). To avoid this solution, we use linear predictivity in the opposite direction [penalized RNN → reference RNN] as the similarity penalty. This asymmetry between predictivity and predictability in their sensitivity to irrelevant dynamics has also been noted in recent work on latent variable models of neural activity (Versteeg et al., 2024; Dabholkar & Barak, 2025). We remark that canonical correlation analysis (Hotelling, 1936; Raghu et al., 2017) can also avoid this undesirable solution, although the extra whitening step incurs a slight additional computational cost.

We define linear predictivity as $r^2(\boldsymbol{X}, \boldsymbol{Y}) = 1 - \min_{\boldsymbol{M} \in \mathbb{R}^{N \times N}} \frac{\|\boldsymbol{X}\boldsymbol{M} - \boldsymbol{Y}\|^2}{\|\boldsymbol{Y}\|^2} = \frac{\|\boldsymbol{U}_X \boldsymbol{Y}\|^2}{\|\boldsymbol{Y}\|^2}$ where $\boldsymbol{U}_X = \boldsymbol{X}(\boldsymbol{X}^\top \boldsymbol{X})^+ \boldsymbol{X}^\top \in \mathbb{R}^{(BL_t) \times (BL_t)}$ projects to the column space of $\boldsymbol{X}$. As the input matrices are

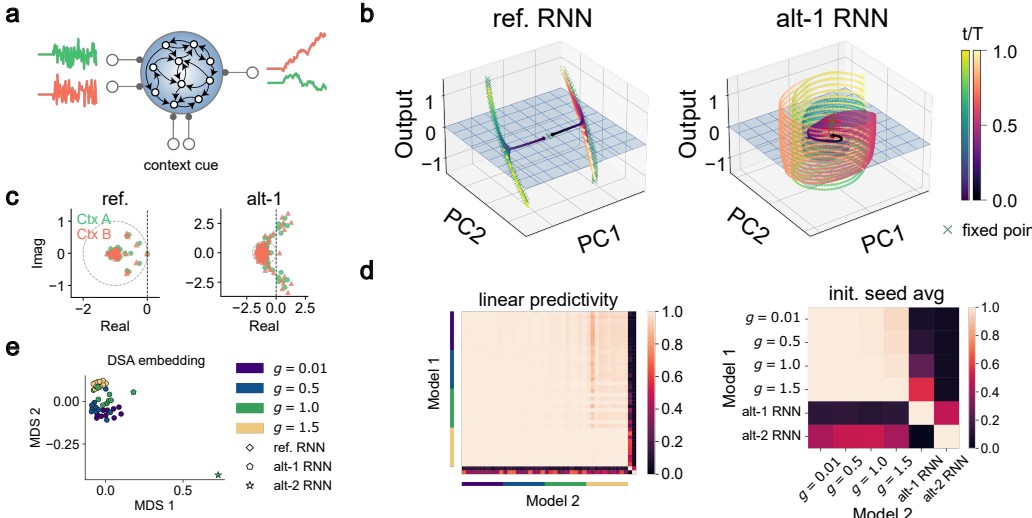

Figure 1: **Similarity-penalized RNNs yield distinct solutions to context-dependent integration.**
**a.** Task schematic: two noisy stimuli are passed as input. In each trial, only the input stream selected by the context cue needs to be integrated, while the other is ignored. **b.** Example trajectories shown along the first two PCs and the output axis for the reference RNN (left) and alt-1 RNN (right), respectively. Trajectories are colored by time and relevant context (Ctx A: viridis, Ctx B: magma) during the corresponding trial. Fixed points (green x's) and unstable oscillatory leading eigenmodes (red bars) are shown. **c.** Representative examples of eigenvalue spectrums for Jacobians computed at fixed points found for the reference RNN (left) and alt-1 RNN (right). **d.** Left: Linear predictivity matrix across RNNs at different initialization scales and seeds trained on the task, along with the alt-1 and alt-2 RNNs. Right: same, but with scores for the standard RNNs averaged over initialization seed. **e.** MDS embedding of the DSA dissimilarity matrix computed across the same RNNs as in **d**.

often rank-deficient in our usage, for numerical stability, we also add a small ridge regularizer when computing the similarity penalty: $S(\boldsymbol{X}, \boldsymbol{Y}) = \frac{\|\boldsymbol{U}_{X,\rho}\boldsymbol{Y}\|^2}{\|\boldsymbol{Y}\|^2}$, where $\boldsymbol{U}_{X,\rho} = \boldsymbol{X}(\boldsymbol{X}^\top\boldsymbol{X} + \rho\boldsymbol{I})^{-1}\boldsymbol{X}^\top$.

## 2.3 Dynamical systems analysis

We probe the dynamical properties of task solutions via numerically solving for fixed points, as in (Sussillo & Barak, 2013). In line with previous studies (Sussillo & Barak, 2013; Maheswaranathan et al., 2019; Driscoll et al., 2024; Kurtkaya et al., 2025), we include approximate fixed points, also referred to as slow points. Where relevant, we also report the stability, eigenvalue spectrum and leading eigenmode(s) that govern the linearized dynamics in the vicinity of each fixed point.

## 3 Results

We analyze and compare similarity-penalized solutions across three neuroscience-style tasks that have been well studied in the literature (Barak et al., 2013; Mante et al., 2013; Maheswaranathan et al., 2019; Schuessler et al., 2020; Smith et al., 2021; Krause et al., 2022; Valente et al., 2022; Costacurta et al., 2024; Driscoll et al., 2024; Huang et al., 2025; Pagan et al., 2025). These tasks span context-dependent processing, discrete and analog memory, and delayed output production. Each of these tasks is associated with a prototypical solution that has been reported across multiple studies, which we briefly describe for each task. Task parameters are specified in A.3.

**Context-dependent integration.** We begin by studying RNNs trained on context-dependent integration (Fig. 1a). For this task, the network receives two streams of noisy input stimuli and a fixed context cue. For a short duration $T_{\text{pre}}$, only the one-hot encoded context cue is shown. Thereafter, the context cue remains on, while the noisy input stimuli are sampled independently at each timestep from $\mathcal{N}(\mu_i, \sigma^2/dt)$ (following the convention in (Mante et al., 2013; Schuessler et al., 2024)). For each trial, the stimuli coherences $\mu_i$ are sampled from $\mathcal{U}[-\mu_{max}, \mu_{max}]$. At each timestep, the

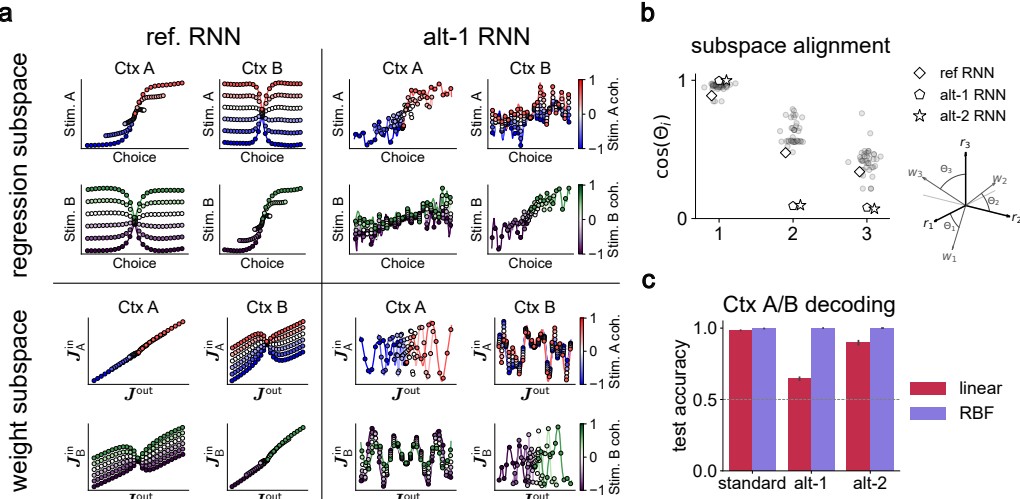

Figure 2: **Linear encoding of task-relevant information is degraded in similarity-penalized RNNs.** Task: context-dependent integration. **a.** Averaged trajectories plotted on different sets of axes, colored by the coherences of the input stimuli. Top row: axes directions estimated via predicting current target output (choice), stimulus A coherence, and stimulus B coherence via linear regression over neural activity aggregated over trials and time points. Bottom row: same averaged trajectories, but plotted on axes of the input and output weights. In each quadrant, left and right plots correspond to context A and B trials, respectively. Colorbars are normalized so that $\pm 1$ corresponds to the minimum/maximum coherence value. **b.** Alignment between the regression and weight subspaces, as measured by the cosine of the principal angles. Grey dots represent alignments computed for the population of standard RNNs. **c.** Decodability of the relevant context from neural activity under linear or RBF kernel regression, as quantified by test accuracy on a heldout set. Error bars report the standard error of the mean. The grey dotted line represents the baseline accuracy.

network must output the cumulative sum (scaled by $dt$) of all inputs received so far in the stimulus channel selected by the context cue. RNNs trained on this task and its binary decision making variant have consistently been found to learn two lines of fixed points (line attractors), one for integrating the relevant stimulus in each context (Mante et al. (2013); Maheswaranathan et al. (2019); Smith et al. (2021); Krause et al. (2022); Pagan et al. (2025)).

To assess the properties of solutions, as in (Maheswaranathan et al., 2019), we first probe all trained networks using task trials of varying stimuli coherences, turning off stimuli noise for visual clarity. In line with previous findings, we observed that all standard RNNs found the aforementioned prototypical solution, regardless of initialization scale and training seed. We illustrate this solution for a reference RNN in Fig. 1b (left), showing activity trajectories plotted on the axes of the first two principal components and the readout. During the context-only period, trajectories quickly segregate into separate regions of state space. Then, in each context, activity is driven along a line of approximate fixed points that densely tile the span of trajectories observed in that context. In contrast, similarity penalized RNNs yielded solutions characterized by oscillatory dynamics (Fig. 1, right). Activity in each context was readily distinguishable by the shape of trajectories, rather than the portion of state space they occupy. Moreover, activity was no longer driven along slow/fixed points. Instead, unstable fixed points with oscillatory eigenmodes were found, but were not used (at least directly) for remembering the cumulative input in either context. Comparing the eigenspectrums of the Jacobians at representative fixed points for both networks confirmed that marginally stable linearized dynamics were only present for fixed points of the reference RNN (Fig. 1c). For brevity, we defer the trajectory and eigenspectrum plots for the alt-2 RNN to the Appendix (Fig. A.1).

We compute linear predictivity scores in both directions between all pairs of models, including the population of standard RNNs and models produced by two iterations of INSD. We find that the representations used by standard RNNs are all highly linearly predictive of each other, with only slight deviations from perfect predictivity observed when predicting models of high initialization scale from models of lower initialization scale (Fig. 1d). Further, similarity penalized RNNs were markedly less predictive and less predictable with respect to standard solutions. To quantify rela-

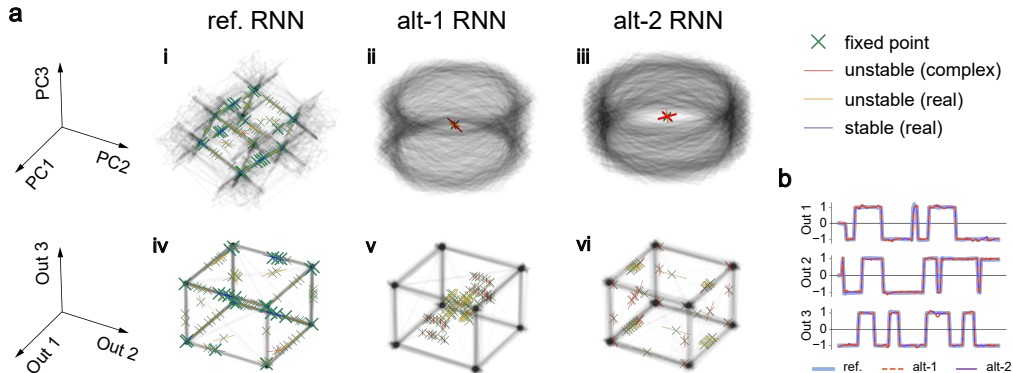

Figure 3: **Sustaining discrete memory states without fixed-point attractors.** Task: 3-bit flipflop. **a.** Example trajectories plotted on the principal component (**i, ii, iii**) and output (**iv, v, vi**) axes, shown for the reference (**i, iv**), alt-1 (**ii, v**), and alt-2 (**iii, vi**) RNNs . Fixed points (green x's) and their leading eigenmodes (colored bars) are shown. A larger marker size is used for stable fixed points. **b.** Example timeseries of network output for all three networks.

tionships between the solutions beyond geometrical similarity, we also compute their Dynamical Similarity Analysis (DSA, Ostrow et al. (2023)) dissimilarity matrix, visualizing the scores via a multi-dimensional scaling embedding (Fig. 1e). This embedding reveals a degree of clustering by initialization scale. However, similarity-penalized solutions achieve a dynamical dissimilarity with respect to the standard population that far exceeds the scale of variability observed across clusters.

Next, we analyzed population responses via projecting activity trajectories onto task-relevant sub-spaces. For the reference and alt-1 RNNs, we first construct a regression-based subspace comprising of the "stimulus A", "stimulus B", and "choice" axes. These directions were estimated via linearly regressing the coherences of stimulus A, stimulus B, and the task target, respectively, from neural activity aggregated across timesteps and 5000 trials. Consistent with prior studies (Mante et al., 2013; Smith et al., 2021; Pagan et al., 2025), projecting the averaged trajectories of standard RNNs onto this set of axes revealed a temporally stable and consistent encoding of the coherences of both input stimuli, regardless of the selected context (Fig. 2a, top left). In contrast, for the alt-1 RNN, the coherence of the relevant stimulus in each trial could still be linearly decoded somewhat consistently, but estimates of the irrelevant stimulus were often inconsistent with actual trial conditions (Fig. 2a, top right). We repeated these analyses, but for a weight-based subspace, projecting averaged trajectories onto the axes $[\boldsymbol{J}_A^{\text{in}}, \boldsymbol{J}_B^{\text{in}}, \boldsymbol{J}^{\text{out}}]$ defined by the input and output weights of each RNN. We again find that, for the standard solution, stimuli coherences for both relevant and irrelevant stimuli can be stably distinguished under these axes (Fig. 2a, bottom left). However, for the alt-1 RNN, the directions encoded by the input weights poorly captured the coherences of both stimuli, regardless of context (Fig. 2a, bottom right). To assess the relationship between the weight and regression subspaces, we quantified their alignment via computing the principal angles between them (Fig. 2b). Across all models, the leading overlap was near unity, likely due to the high alignment between the regression "choice" axis and $\boldsymbol{J}^{\text{out}}$ weight axis. Although the standard RNNs demonstrated varying degrees of moderate alignment between the remaining axes, these angles were near orthogonal for both the alt-1 and alt-2 RNNs. Finally, we assessed the extent to which task context—the most basic task variable—can be accurately decoded from activity. Consistent with the geometric picture of Fig. 1b, we find that context is linearly decodable at high accuracy for standard RNNs, whereas the alt-1 RNN (and to a lesser extent, alt-2 RNN) requires additional nonlinear featurization of representations for context to be decodable at similarly high accuracy (Fig. 2c).

**3-bit flipflop.** We next seek alternative solutions on 3-bit flipflop, a simple discrete memory task. For this task, three input channels are given. At each timestep, each channel independently has a probability $p$ of having an upward or downward spike of magnitude $1/dt$, with both directions having equal probability. The target output for the network begins at 0 for all channels, and thereafter tracks the sign of the last spike in each channel. Trained RNNs consistently learn the most minimal and sensible solution: fixed point attractors arranged in a cube associated with each of the 8 main output states (aside from the starting outputs at 0), as well as saddle points whose unstable directions are aligned with edges of the cube to facilitate state transitions (Barak et al., 2013; Maheswaranathan

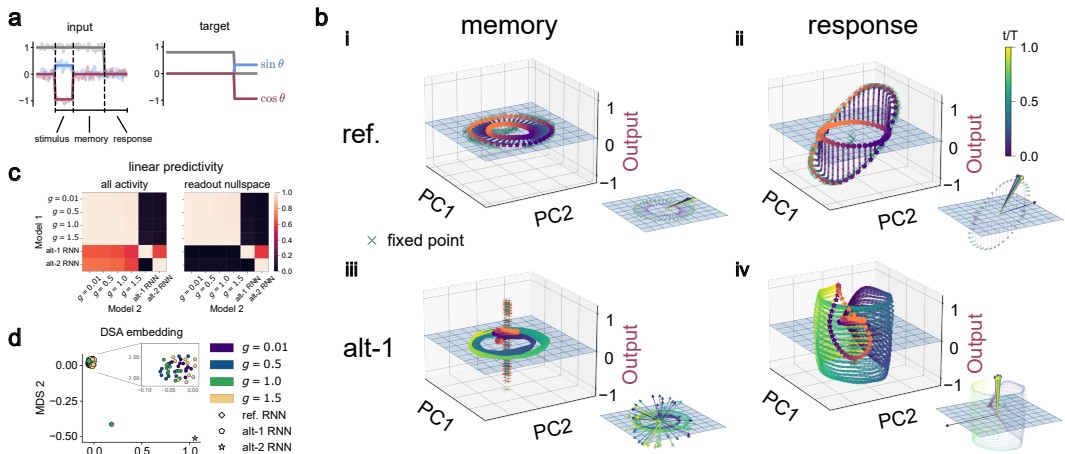

Figure 4: **Similarity-penalized RNNs find dynamic, rather than persistent, encoding of analog memories. a.** Task: MemoryPro. an angle encoded as a 2D vector is passed as input during a stimulus phase, followed by a memory phase where the angle input is absent. Only once the fixation cue (grey) is removed must the network output the angle that was observed. All inputs are noisy. **b.** Example trajectories divided by the memory (**i, iii**) and response (**ii, iv**) phases, shown for the reference RNN (**i, ii**) and alt-1 RNN (**iii, iv**). All plots are along the first two memory phase PCs and the cos output axis, with trajectories colored by time. The start and end of every trajectory is colored by the target output angle. Fixed points (green x's) and unstable oscillatory eigenmodes (red bars) are shown. The right corner of each subplot shows the direction encoding the target angle over time, as estimated by linear regression. **c.** Linear predictivity matrices comparing standard RNNs at different initialization scales to similarity penalized RNNs (alt-1, alt-2). Scores involving standard RNNs are averaged with respect to the initialization seed. Left: base linear predictivity scores. Right: linear predictivity scores when activity is first projected to the readout nullspace. **d.** MDS embedding of the DSA similarity matrix computed across the same RNNs as in **c**.

et al., 2019; Ostrow et al., 2023). We plot trajectories of solutions as well as fixed points for a reference, alt-1 and alt-2 RNN trained on this task. We confirm that the reference RNN indeed learns the standard solution involving the cube of stable fixed points, and saddle points that transition between them (Fig. 3a,i). Moreover, the geometrical structure of activity in PCA space is minimal in the sense that it mirrors the cube-like geometry of the task output. For the similarity penalized RNNs, however, observed trajectories no longer show this geometry in PCA space, and instead follow oscillations generated by unstable fixed points with complex leading eigenmodes (Fig. 3a,ii,iii).

Despite these apparent differences in representational geometry, all three networks must ultimately produce cube-like geometry when trajectories are projected onto the output subspace; this is demanded by the structure of the target output of the task. Thus, to compare the solutions found more aptly, we also plot trajectories and fixed points on the output axes of each RNN. This reveals that, even in the output subspace, similarity-penalized RNNs exhibit distinct arrangements and stability properties of fixed points. In this example, the alt-1 RNN lacks fixed points that stabilize any of the output states, instead showing two groups of unstable fixed points with oscillatory eigenmodes, and saddle points that appear to transition between them (Fig. 3a,v). The alt-2 RNN recovers the presence of fixed points at each corner, but they are no longer stable/attractive (Fig. 3,vi). Moreover, the directions of saddle points that line the edges of the cube are often misaligned. These differences in dynamical motifs manifest as slight but noticeable imperfections in the output produced by the similarity-penalized RNNs (Fig. 3b). We also assess the similarity of representations across all RNNs using linear predictivity and DSA (Fig. A.2). Similar to the findings for context-dependent integration, all standard solutions are found to be perfectly linear predictive of each other, whereas the similarity-penalized RNNs occupy disparate areas of the DSA MDS embedding.

**MemoryPro.** Lastly, we turn our attention to the MemoryPro task (Fig. 4a). The RNN receives three piecewise constant inputs: a fixation cue and 2 stimuli channels encoding an angle. For each trial, the angle $\theta$ is sampled from $\mathcal{U}[-\pi, \pi]$. Following Driscoll et al. (2024), at train time, stimuli and response onsets and offsets are variable. Specifically, after a delay of length $T_{\text{del}} \sim \mathcal{U}[T_{\text{del}}^-, T_{\text{del}}^+]$,

Figure 5: **Similarity penalized models can outperform standard models in difficult task regimes. a, b.** Mean squared error on the context-dependent integration (**a**) and flipflop (**b**) tasks, respectively, across different noise scales $\sigma$ and trial length scaling $T_{\text{mult}}$. The grey dotted lines indicate the noise scale used during training. **c.** Accuracy on the MemoryPro task versus fractional memory load, at different noise scales $\sigma$. We assess accuracy using the same criteria as in (Driscoll et al., 2024). Small blue dots represent the scores achieved by the population of standard RNNs.

the angle stimuli $(\sin\theta \quad \cos\theta)^\top$ are shown for a duration $T_{\text{stim}} \sim \mathcal{U}[T_{\text{stim}}^-, T_{\text{stim}}^+]$. Then, the stimuli are turned off for a duration $T_{\text{mem}} \sim \mathcal{U}[T_{\text{mem}}^-, T_{\text{mem}}^+]$, following which the fixation cue is removed and the response period begins. For a duration $T_{\text{resp}} \sim \mathcal{U}[T_{\text{resp}}^-, T_{\text{resp}}^+]$, the network must output the angle seen during the stimuli phase, also as a 2D vector. The network must also produce an output that tracks the fixation cue. All three inputs are also subjected to independent noise at each timestep, drawn from $\mathcal{N}(0, \sigma^2)$. Previous studies consistently report the following prototypical solution: during the memory phase, angles are encoded along a ring manifold of persistent states in the output nullspace, stabilized by a ring attractor. During the response phase, this ring of fixed points quickly rotates to become output potent (Driscoll et al., 2024; Costacurta et al., 2024; Hazelden et al., 2025).

To probe the properties of solutions, we plot trajectories collected over trials with various target angles and in the absence of input noise (Fig. 4b). As in Driscoll et al. (2024), we use the axes of the first two memory phase PCs and the $\cos\theta$ output channel, separating trajectories by the memory and response phases. Our results confirm that standard RNNs ubiquitously find the prototypical solution involving a ring attractor that rotates outwards, shown for the reference network (Fig. 4b,i,ii). We also plot the direction in activity space that best predicts the target angle via linear regression at each timestep, confirming that memorized angles are statically encoded (Fig. 4b,i,ii, bottom right). In contrast, the alt-1 RNN exhibits rotational dynamics during the memory phase that nonetheless maintains the relative ordering of trajectories by their corresponding target output (Fig. 4b, iii). The ring of fixed points is no longer present, and is instead replaced by a line of unstable fixed points with oscillatory leading eigenmodes. Linear decoding analysis reveals that the direction encoding the target angle is indeed rotating with the activity (Fig. 4b, iii, bottom right). Moreover, this direction even acquires output potency at times, despite the fact that the output potent component is, by task necessity, a low-variance fraction of the activity during the memory phase. During the response phase, these trajectories continue to oscillate, but rotate to become output potent (Fig. 4b, iv). We defer the corresponding plots for the alt-2 RNN to the Appendix (Fig. A.3).

As done for previous tasks, we compute linear predictivity and DSA dissimilarity scores between all pairs of models across the standard and similarity-penalized RNNs. While the linear predictivity of standard solutions from similarity-penalized solutions is degraded, we find that it is still significantly above zero (Fig. 4c, left). However, this partial predictivity is ablated once activities are projected into their respective readout nullspaces. This indicates that the only component of activity that the similarity penalized models can predict from standard solutions is that which is necessary to solve the task, namely, the output potent component. An MDS embedding of the DSA dissimilarity matrix confirms that the similarity-penalized RNNs achieve dynamically dissimilar solutions (Fig. 4d).

**Assessing solutions by their performance under atypical task conditions.** Across all three tasks, we found solutions that appear distinct from standard solutions by a variety of measures. However, are these solutions actually functionally distinct, or are they merely approximating the standard solution in ways that are difficult to discern? To answer this question, we tested all models under task conditions seldom or never seen during training. For the context-dependent integration task, we measured task performance across different noise scales $\sigma$ of the input stimuli. We also introduce and sweep over the parameter $T_{\text{mult}}$, a factor that uniformly scales the duration of trials. We conduct a similar performance sweep for the flipflop task. For the MemoryPro task, we again sweep the

input noise scale $\sigma$, but also sweep the fractional memory load, which we define as $\frac{T_{\text{mem}}}{T_{\text{stim}}+T_{\text{mem}}}$. To tune this parameter, we fix the duration of the pre-stimulus and response phases, as well as the total duration of the stimulus and memory phases combined. We then adjust the timing of the transition from the stimulus to the memory phase to produce test trials of varying fractional memory loads.

We report model performance across these sweeps in Fig. 5, as well as the corresponding effective dimensionality of activity as measured by the participation ratio in Fig. A.4. Across all tasks, we find that standard RNNs typically outperform similarity-penalized RNNs when tested under conditions seen during training (Fig. 5). However, we also observe many cases where similarity-penalized RNNs outperform standard models. For instance, for the context-dependent integration task, the alt-2 RNN moderately outperforms the population of standard RNNs in highly noisy conditions, all the while remaining robust to lengthened trial durations. For the flipflop task, although we observe near-identical performance across most models, the alt-2 RNN achieves a moderate but significant gain in relative performance when noise is high. Lastly, for MemoryPro, we observe that the alt-1 RNN significantly outperforms standard RNNs on the most difficult trials, where both noise and memory load are high, but significantly underperforms the population under low memory loads. The alt-2 RNN only matched or underperformed the population under all conditions, suggesting that it simply failed to learn the task as well. Altogether, these performance deviations confirm that similarity-penalized models indeed produce solutions that are functionally distinct.

## 4 DISCUSSION

Generating a rich set of diverse hypotheses that can be tested against experimental data is foundational for progressing our understanding of the brain. Motivated by recent observations of dynamic collapse in task-trained RNNs (Maheswaranathan et al., 2019; Driscoll et al., 2024; Hazelden et al., 2025), we propose a method called Iterative Neural Similarity Deflation (INSD) for expanding the space of accessible solutions. Across three neuroscience-style tasks, we extensively study and compare the solutions generated by iteratively penalizing the linear predictivity of past solutions. These analyses revealed alternative solutions that did not directly use simple dynamical motifs such as fixed point attractors or continuous slow manifolds to store information. Instead, similarity-penalized RNNs tended to produce activity characterized by quasi-periodic oscillatory modes. Further analysis revealed that these oscillations were not simply nuisance dynamics that emerged as a peculiarity of the similarity penalty, but rather actively supported the dynamic encoding of task-relevant information. These solutions are reminiscent of a theory proposed by Park et al. (2023) on how memories can be stably maintained in the phase difference between two oscillations, rather than through persistent attractor states. In the same vein, recent work by Ritter & Chadwick (2025) argues that optimally efficient and noise-robust working memory requires high-dimensional rotational dynamics, and further finds signatures of such dynamics in monkey prefrontal cortex. These observations are consistent with our finding of improved robustness for some similarity-penalized solutions.

For context-dependent integration, unlike similarity-penalized RNNs, standard RNNs produced solutions where task-relevant information was stably represented in linear subspaces, consistent with neural data recorded during analogous tasks (Mante et al., 2013; Pagan et al., 2025). Thus, a natural concern is that similarity-penalized RNNs may produce solutions whose population coding properties are not realistic. However, we argue that being able to also find unrealistic solutions is crucial for probing when and why simple solutions align with biology. Moreover, in principle, one could construct networks that interpolate between standard and similarity-penalized solutions. Most simply, this could be achieved by an RNN with two populations of neurons, one dedicated to implementing each solution. Much as how ensembling is used in machine learning to reduce variance and improve generalization, such mixed models may possibly enjoy greater robustness, all the while maintaining more realistic linear encoding properties at the population level. We leave a more detailed investigation of this idea to future work.

Finally, we acknowledge that linear predictivity is an imperfect measure of both dynamical similarity and functional equivalence (Ostrow et al., 2023; Qian et al., 2024; Braun et al., 2025). The recently proposed Dynamical Similarity Analysis (DSA, Ostrow et al. (2023)) has been shown to effectively identify RNN solutions whose dynamical properties are only superficially distinct, while other metrics often fall short. However, computing this metric as a similarity penalty in an online fashion would be prohibitively computationally expensive. Despite the limitations of linear predictivity, we

found that penalizing the predictivity of representations used by standard RNNs was sufficient to generate solutions with distinct dynamical features and unique task performance profiles. While we did not perform an extensive sweep over the strength of the predictivity penalty here, solutions that differ from standard solutions in more fine-grained ways may potentially also be discoverable if the penalty is set to be small. For instance, for context-dependent decision making, a more granular form of solution degeneracy can be characterized in terms of the relative arrangements of input vectors and eigenvectors of the linearized dynamics across contexts (Pagan et al., 2025). Since the penalty we use relates to neural activity, our approach likely cannot fully capture degeneracies in the relative arrangements between left eigenvectors and input vectors, but may be able to explore degeneracies in the relative arrangements of the right eigenvectors (integration directions) across contexts, which do directly manifest in activity.

A limitation of our study is that we focus on simple single-task settings where standard solutions invoke attractor dynamics. Future work should investigate tasks that require transient dynamics, such as timing tasks, where standard RNN solutions are already somewhat varied (Turner et al., 2021; Beiran et al., 2023; Huang et al., 2025). Experiments in multitask settings would also be insightful for understanding whether greater task demands make it more difficult to find solutions that are not linearly predictive of reference solutions (Cao & Yamins, 2024; Huang et al., 2025). More generally, this "capacity" for distinct solutions is expected to depend on task difficulty and network size, and could be systematically explored using INSD. Such experiments could serve as a way to empirically assess ideas such as the Contravariance Principle and the Platonic Representation Hypothesis (Cao & Yamins, 2024; Huh et al., 2024), both of which assert that sufficiently difficult or constraining task demands necessitate representational convergence across systems.

## ACKNOWLEDGMENTS

We thank Jacob A. Zavatone-Veth and David G. Clark for insightful discussions and comments on a previous version of this manuscript. W.Q. is supported by a Kempner Graduate Fellowship. C.P. is supported by an NSF CAREER Award (IIS-2239780), DARPA grants DIAL-FP-038 and AIQ-HR00112520041, the Simons Collaboration on the Physics of Learning and Neural Computation, and the William F. Milton Fund from Harvard University. This work has been made possible in part by a gift from the Chan Zuckerberg Initiative Foundation to establish the Kempner Institute for the Study of Natural and Artificial Intelligence.

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

# A APPENDIX

## A.1 TRAINING AND OTHER MISCELLANEOUS DETAILS

For all experiments, we use RNNs with $N = 128$ neurons. All RNNs are trained in PyTorch. We use the Adam optimizer with a learning rate of $10^{-3}$, a weight decay of $10^{-5}$, and a batch size of 32. For the strength of the similarity penalty, we use $\lambda = 0.05$ throughout. When computing linear predictivity, we use $\rho = 10^{-3}$ as the ridge regularizer. RNNs trained as part of the INSD procedure are initialized at the scale $g = 1$. All networks are trained for a minimum of $10^6$ iterations, with training terminating when the loss stops improving. Training runs were primarily done using 4th Generation Intel Xeon CPUs; GPU acceleration was not necessary.

For computing DSA dissimilarity matrices, we use the open source package from (Ostrow et al., 2023). Across all tasks, we used a rank of 100, 8 delays, and a delay interval of 10 timesteps. The delay parameters were selected to be compatible with trials of duration 100 timesteps, as used for context-dependent integration and 3-bit flipflop.

For finding fixed points, we use the open source package FixedPointFinder (Golub & Sussillo, 2018). We report approximate fixed points with velocities $q$ spanning $q = 5 \times 10^{-4}$ to $q = 10^{-9}$, and subsample redundant fixed points by adjusting the uniqueness tolerance parameter. As in (Driscoll et al., 2024), we report fixed points over a wide range of velocity tolerances to best account for variations in relevant timescales across the different tasks.

Training and analysis code is publicly available at `https://github.com/wqian0/INSD`.

## A.2 A BRIEF NOTE ON NEURAL SIMILARITY PENALTY LOOPHOLES

We model the scenario described in the main text as follows: we are given two sets of neural representations $\boldsymbol{X}, \boldsymbol{Y} \in \mathbb{R}^{P \times N}$. Suppose that the representations in $\boldsymbol{X}$ are contained in a low dimensional subspace of dimension $k \ll N, P$. We represent this by factorizing $\boldsymbol{X} = \boldsymbol{L}\boldsymbol{W}$, where $\boldsymbol{L} \in \mathbb{R}^{P \times k}$ are the latent representations and $\boldsymbol{W} \in \mathbb{R}^{k \times N}$. Suppose further that $\boldsymbol{Y}$ is composed of identical latents, along with some irrelevant noise in other dimensions. We write this as $\boldsymbol{Y} = [\boldsymbol{L}\boldsymbol{Q} \quad \sigma\boldsymbol{Z}]$, where $\boldsymbol{Q} \in \mathbb{R}^{k \times k}$ is an orthogonal matrix, $\boldsymbol{Z} \in \mathbb{R}^{P \times d}$ represents the irrelevant noise, and $d = N - k$. For simplicity, we model the entries of $\boldsymbol{Z}$ as drawn i.i.d from $\mathcal{N}(0, 1)$. Below, we compute and describe the behavior of various similarity metrics on these inputs at large $N$, $P$, and $\sigma$.

### A.2.1 CENTERED KERNEL ALIGNMENT (CKA)

We focus on linear CKA:

$$\text{CKA}(\boldsymbol{X}, \boldsymbol{Y}) = \frac{\|\boldsymbol{X}^\top \boldsymbol{Y}\|^2}{\|\boldsymbol{X}^\top \boldsymbol{X}\|\|\boldsymbol{Y}^\top \boldsymbol{Y}\|} \tag{5}$$

We expand the numerator as $\|\boldsymbol{X}^\top \boldsymbol{Y}\|^2 = \|\boldsymbol{X}^\top \boldsymbol{L}\boldsymbol{Q}\|^2 + \sigma^2 \|\boldsymbol{X}^\top \boldsymbol{Z}\|^2$.

We also expand $\|\boldsymbol{Y}^\top \boldsymbol{Y}\|^2 = \|\boldsymbol{L}^\top \boldsymbol{L}\|^2 + 2\sigma^2 \|\boldsymbol{L}^\top \boldsymbol{Z}\|^2 + \sigma^4 \|\boldsymbol{Z}^\top \boldsymbol{Z}\|^2$.

At large $N$, we can approximate $\boldsymbol{Z}\boldsymbol{Z}^\top/d \to \boldsymbol{I}_P$. This allows the simplification $\|\boldsymbol{X}^\top \boldsymbol{Z}\|^2 = \text{Tr}(\boldsymbol{Z}^\top \boldsymbol{X}\boldsymbol{X}^\top \boldsymbol{Z}) = \text{Tr}(\boldsymbol{Z}\boldsymbol{Z}^\top \boldsymbol{X}\boldsymbol{X}^\top) = d\|\boldsymbol{X}\|^2$, and $\|\boldsymbol{Z}^\top \boldsymbol{Z}\|^2 = d^2 P$. At large $\sigma$, we can drop subleading terms in $\sigma$, giving

$$\text{CKA}(\boldsymbol{X}, \boldsymbol{Y}) \approx \frac{\sigma^2 d\|\boldsymbol{X}\|^2}{\sigma^2 d\sqrt{P}\|\boldsymbol{X}^\top \boldsymbol{X}\|} \leq \mathcal{O}\left(\sqrt{\frac{k}{P}}\right), \tag{6}$$

where the final inequality follows from the bound $\|\boldsymbol{X}\|^2 \leq \sqrt{k}\|\boldsymbol{X}^\top \boldsymbol{X}\|$.

Thus, CKA between otherwise identical representations can be suppressed through irrelevant noise.

### A.2.2 REPRESENTATIONAL SIMILARITY ANALYSIS (RSA)

We take RSA to refer to the cosine similarity between the squared Euclidean distance representational dissimilarity matrices (RDMs), as in Williams (2024).

Let $\boldsymbol{D}^X_{ij} = \|\boldsymbol{x}_i - \boldsymbol{x}_j\|^2$ and $\boldsymbol{D}^Y_{ij} = \|\boldsymbol{y}_i - \boldsymbol{y}_j\|^2$ represent the $P \times P$ RDMs. We have:

$$\text{RSA}(\boldsymbol{X}, \boldsymbol{Y}) = \frac{\langle \boldsymbol{D}^X, \boldsymbol{D}^Y \rangle}{\|\boldsymbol{D}^X\|\|\boldsymbol{D}^Y\|} \tag{7}$$

We can write $\boldsymbol{D}^Y_{ij} = \boldsymbol{D}^S_{ij} + \sigma^2 \boldsymbol{D}^Z_{ij}$, where $\boldsymbol{S} = \boldsymbol{L}\boldsymbol{Q}$. Dropping terms subleading in $\sigma$, we have

$$\text{RSA}(\boldsymbol{X}, \boldsymbol{Y}) \approx \frac{\langle \boldsymbol{D}^X, \boldsymbol{D}^Z \rangle}{\|\boldsymbol{D}^X\|\|\boldsymbol{D}^Z\|} \tag{8}$$

Note that $\mathbb{E}[\boldsymbol{D}^Z_{ij}] = 2d$ for $i \neq j$. At large $N$ (and therefore large $d$), we can expect concentration, yielding $\boldsymbol{D}^Z/d \to 2(\boldsymbol{J} - \boldsymbol{I})$, where $\boldsymbol{J}$ is a $P \times P$ matrix of ones. Thus, we have

$$\text{RSA}(\boldsymbol{X}, \boldsymbol{Y}) \approx \frac{\sum_{i \neq j} \boldsymbol{D}^X_{ij}}{\sqrt{\sum_{i \neq j}(\boldsymbol{D}^X_{ij})^2}\sqrt{P(P-1)}} = \mathcal{O}\left(\frac{1}{P}\right). \tag{9}$$

Thus, RSA is also suppressed by irrelevant noise.

### A.2.3   LINEAR PREDICTIVITY [REF. $\to$ PENALIZED]

As in the main text, define the projection operator $\boldsymbol{U}_X = \boldsymbol{X}(\boldsymbol{X}^\top \boldsymbol{X})^+ \boldsymbol{X}^\top$. We have that

$$r^2(\boldsymbol{X}, \boldsymbol{Y}) = \frac{\|\boldsymbol{U}_X \boldsymbol{Y}\|^2}{\|\boldsymbol{Y}\|^2}. \tag{10}$$

We can write $\|\boldsymbol{U}_X \boldsymbol{Y}\|^2 = \|\boldsymbol{S}\|^2 + \sigma^2 \|\boldsymbol{U}_X \boldsymbol{Z}\|^2$, where we have used that $\boldsymbol{U}_X \boldsymbol{S} = \boldsymbol{S}$, as by construction, $\boldsymbol{S} = \boldsymbol{L}\boldsymbol{Q}$ is contained in the column space of $\boldsymbol{X} = \boldsymbol{L}\boldsymbol{W}$. Similarly, we have $\|\boldsymbol{Y}\|^2 = \|\boldsymbol{S}\|^2 + \sigma^2 \|\boldsymbol{Z}\|^2$, yielding $r^2(\boldsymbol{X}, \boldsymbol{Y}) \approx \frac{\|\boldsymbol{U}_X \boldsymbol{Z}\|^2}{\|\boldsymbol{Z}\|^2}$ at large $\sigma$. Finally, at large $N$, we have that

$$r^2(\boldsymbol{X}, \boldsymbol{Y}) \approx \frac{\|\boldsymbol{U}_X \boldsymbol{Z}\|^2}{\|\boldsymbol{Z}\|^2} \to \frac{\mathbb{E}\left[\|\boldsymbol{U}_X \boldsymbol{Z}\|^2\right]}{\mathbb{E}\left[\|\boldsymbol{Z}\|^2\right]} = \frac{kd}{Pd} = \frac{k}{P}, \tag{11}$$

demonstrating that linear predictivity in this direction is also suppressed by irrelevant noise.

### A.2.4   LINEAR PREDICTIVITY [PENALIZED $\to$ REF.]

Consider the opposite direction:

$$r^2(\boldsymbol{Y}, \boldsymbol{X}) = \frac{\|\boldsymbol{U}_Y \boldsymbol{X}\|^2}{\|\boldsymbol{X}\|^2}. \tag{12}$$

Since the column space of $\boldsymbol{Y}$ contains that of $\boldsymbol{X}$, we have $\|\boldsymbol{U}_Y \boldsymbol{X}\|^2 = \|\boldsymbol{X}\|^2$, yielding $r^2(\boldsymbol{Y}, \boldsymbol{X}) = 1$. Thus, perfect linear predictivity is maintained.

### A.3   TASK PARAMETERS

**Context-dependent integration:** We use a timestep of $dt = 0.1$, a context-only duration $T_{\text{pre}} = 2.5$ (25 timesteps), and a total trial duration of $T = 10$ (100 timesteps). We set the noise scale to $\sigma = \sqrt{0.1}$.

**3-bit flipflop:** We use a timestep of $dt = 0.2$, and a total trial duration of $T = 20$ (100 timesteps). We set $p = 0.1$ as the spike probability per timestep.

**MemoryPro:** We use a timestep of $dt = 0.2$. Mirroring timing parameters selected in (Driscoll et al., 2024), we set $T^-_{\text{del}} = T^-_{\text{resp}} = 3/dt$, $T^+_{\text{del}} = T^+_{\text{resp}} = 7/dt$, $T^-_{\text{stim}} = T^-_{\text{mem}} = 2/dt$, and $T^+_{\text{stim}} = T^+_{\text{mem}} = 16/dt$. We use a noise scale of $\sigma = 0.1$. As in (Costacurta et al., 2024), we scale down the output channel corresponding to the fixation target by a factor of $0.8$.

## A.4 ADDITIONAL FIGURES

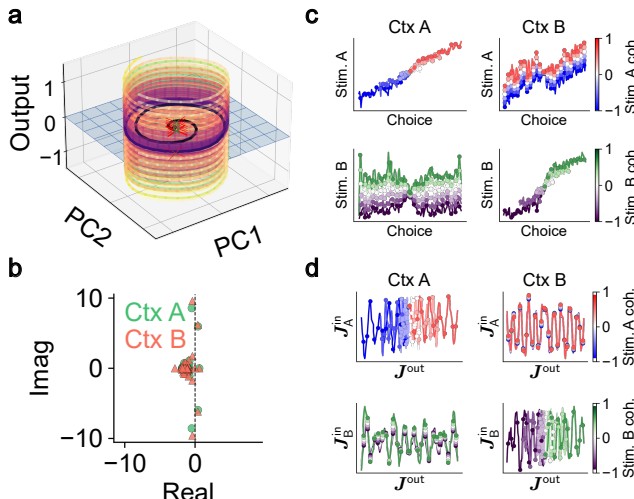

Figure A.1: **Properties of the alt-2 RNN for the context-dependent integration task. a,b.** Analogous to Figs. 1b,c. As for the alt-1 RNN, we observe oscillatory dynamics, as well as fixed points with unstable oscillatory modes. However, these oscillatory modes are of much higher frequency. **c,d.** Analogous to Fig. 2a. Unlike the alt-1 RNN, average trajectories plotted in the regression subspace to some extent maintain the relative ordering of the coherences of both stimuli. This is likely explained by the alt-2 RNN still retaining a degree of linear predictivity of standard RNN representations, something that was entirely absent for the alt-1 RNN (Fig. 1d). However, representations in the weight subspace reveal no consistent representation of stimuli coherences.

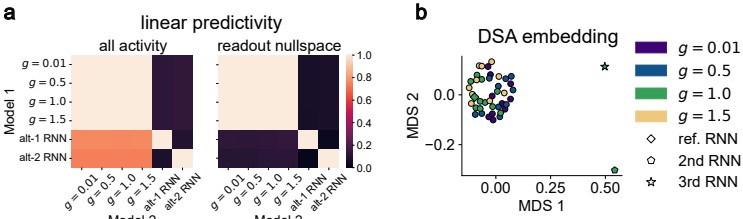

Figure A.2: **Similarity measures across standard and similarity-penalized models trained on the 3-bit flipflop task.** Figures are analogous to those in Fig. 4c,d. Similarity-penalized RNNs retain some degree of linear predictiity of standard RNNs, but that effect is ablated once representations are projected to readout nullspaces. As for other tasks, we also observe a DSA embdding that significantly separates the solutions similarity-penalized RNNs from those found by standard RNNs.

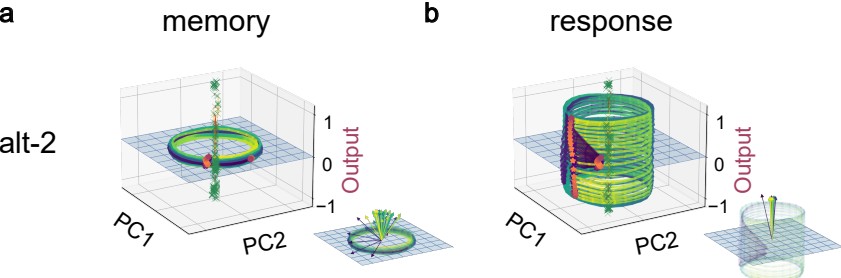

Figure A.3: **Properties of the alt-2 RNN for the MemoryPro task.** Figures are analogous to those in Fig. 4b. We again observe oscillatory dynamics supported by a center of unstable fixed points. This RNN does poorly on the task relative to the RNNs shown in Fig. 4b, as indicated by the activity itself prematurely acquiring significant output potence during the memory phase.

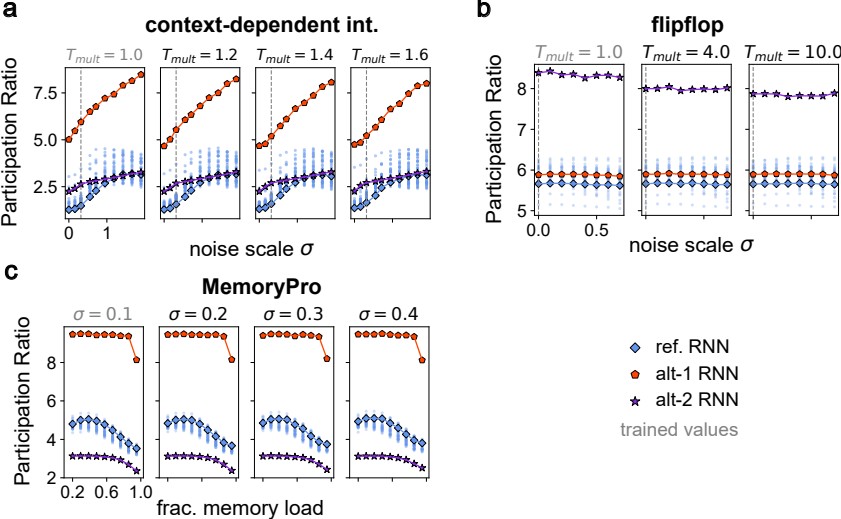

Figure A.4: **Effective dimensionality over different task conditions.** Plots are analogous to those in Fig. 5, but instead report the participation ratio, computed over neural activity during the task.

## A.5 LLM Usage Statement

Large language models were used sparingly for the sole purpose of trimming and polishing text. All technical contributions, experiments, analyses, and figures are the authors' own.

