# OpenReview forum: "Discovering alternative solutions beyond the simplicity bias in recurrent neural networks"
_ICLR.cc/2026/Conference — ICLR 2026 Poster_

### Official Review · Reviewer_657y · 2025-10-29

**Soundness:** 4
**Presentation:** 4
**Contribution:** 4
**Rating:** 8
**Confidence:** 5

**Summary:**

This paper introduces Iterative Neural Similarity Deflation (INSD), a method for overcoming the simplicity bias that often causes recurrent neural networks trained on neuroscience-style tasks to converge to the same fixed-point or low-dimensional dynamical solutions. The authors propose penalizing the linear predictivity between neural activities of new and previously trained networks, effectively forcing networks to explore alternative representational geometries and dynamics. They test INSD on three canonical tasks—context-dependent integration, 3-bit flip-flop, and MemoryPro—and show that it yields qualitatively different solutions characterized by oscillatory or high-dimensional dynamics rather than static attractors. These alternative networks occasionally perform better under challenging, out-of-distribution conditions, suggesting that the INSD procedure can reveal functionally distinct computational strategies.

**Strengths:**

I really liked this paper. There is a notion in computational neuroscience that certain tasks somehow "induce", out of necessity, a particular type of solution in a network. This paper shows that there are many different potential solutions to a particular task.

**Weaknesses:**

The examples are all toy. I would be extremely interested to see how this technique performs in larger-scale settings, such as when training SSMs and/or Transformers on Long Range Arena (or a similarly difficult benchmark).

**Questions:**

1. The paper focuses primarily on relatively simple, neuroscience-style tasks. Is there a specific reason the authors did not extend their analysis to larger or more complex settings, such as multi-task RNNs or higher-dimensional architectures? It would be valuable to understand whether the proposed approach scales effectively beyond toy examples.

2. The authors might consider situating their work in relation to recent theoretical ideas such as the [Contravariance Principle](https://arxiv.org/pdf/2104.01489) and the [Platonic Representation Hypothesis](https://arxiv.org/abs/2405.07987), both of which posit that sufficiently difficult tasks drive convergent representational structures across systems. Could the proposed INSD framework be used to empirically probe these claims—for instance, by testing whether multiple, distinct dynamical solutions can coexist for more complex tasks where such representational convergence is expected?

---

> ### Author Response · Authors · 2025-11-20
> **Response**
>
> We thank the reviewer for their suggestions and comments, and are glad to hear that they liked our work. Below we respond to the two main points raised in Weaknesses and Questions:
>
> ### Choice of tasks for experiments:
> There are a few reasons why we chose to focus on simple neuroscience-style tasks for this paper. Firstly, such tasks are well studied, and are very well-known to the target audience of this work: computational neuroscientists looking to use RNNs as hypothesis generators. Moreover, these tasks represent a setting where the contrast between the solution space degeneracy expected to exist by heuristics like the Contravariance Principle (such simple tasks should not be very constraining!), and the overwhelming uniformity of solutions actually observed and reported across the literature, is most striking. Such simple tasks are also more amenable to being studied in a laboratory setting in a systematic way with neural recordings or  circuit/task perturbations, and thus are a natural choice for testable hypothesis generation.
> Nonetheless, we agree that applying the method in multitask settings and on more complex tasks in general would be insightful--something we have noted as a future direction in the Discussion. Although we focus on neuroscientifically motivated use cases, we also agree that this method is quite general and could also be used to generate improved solution diversity for ML tasks.
>
> ### Contravariance Principle and related concepts:
> We agree that it would be quite interesting to assess the Contravariance Principle and the Platonic Representation Hypothesis in a systematic manner by applying INSD to tasks of increasing complexity. Although we briefly allude to the connection between our work and the Contravariance Principle  at the end of the paper, we agree that it would be useful to make these connections explicit. We have expanded the discussion there accordingly: "More generally...across systems." (lines 496-500).

---

### Official Review · Reviewer_SAv4 · 2025-10-31

**Soundness:** 3
**Presentation:** 4
**Contribution:** 4
**Rating:** 6
**Confidence:** 4

**Summary:**

The authors propose an algorithm for training a population of RNNs on the same task, in a manner that encourages diverse solutions. This is done by adding a cost term that ensures activity in the space orthogonal to the output is orthogonal to previous networks. They show this generates solutions that differ in both geometry and topology.

**Strengths:**

When RNNs are used as hypothesis generators, it is crucial to explore the space of hypotheses. This is usually done in a somewhat heuristic manner, and a more systematic approach is very important.

The idea is novel, original, elegant, and seems to work.

**Weaknesses:**

The other part of hypothesis generation is understanding the mechanism behind each RNN. This part was lacking. How exactly do the alternative RNNs solve the different tasks? In Turner 2021, the mechanism is explained. While this does not always have to be possible, it would be interesting to see a case where it is.

Are the alternatives more or less similar to experimental data?

**Questions:**

Line 148: this is somewhat similar to the argument of bijectivity in latent variable inference (Versteeg et al 2023, Dabholkar et al 2025).

What is the capacity to produce different solutions? How many can there be, and what does this depend on? For instance, Training a 1-bit flip flop can be easily done with a 2-unit GRU. Can the method work already there (or maybe in 3D), thereby allowing phase space analysis?

How robust are these to noise in the actual dynamics, or in connectivity?

---

> ### Author Response · Authors · 2025-11-20
> **Response Part 1**
>
> We thank the reviewer for their useful questions and comments, and appreciate their assessment of the merits of our approach. We respond to points raised under Weaknesses and Questions below. Because of overlap with some concerns raised by reviewer `ReZK`, a few parts of responses there have been repeated here as well.
>
> ### Regarding understanding solutions found by similarity-penalized RNNs:
> We focus the present work on presenting a systematic method for finding alternative solutions, and providing extensive evidence that the solutions it finds are indeed unique when sliced along multiple commonly used levels of understanding (encoding of task-relevant information, arrangements of slow/fixed points, and functional properties). An even more in-depth interpretation of the mechanisms used by the alternative solutions found is also important, but would require significantly more sophisticated analysis due to their complexity (as reasoned later below); this is somewhat secondary to the goal of this paper.
>
> That said, we think it is useful to echo the excellent point raised in the intro of Turner 2021: what exactly constitutes an understanding of a solution? While we agree that a full understanding of the alternative solutions' properties was not quite reached, we argue that we have captured many nontrivial aspects of the solutions that neuroscientists practically would care about, particularly along the same aspects focused on in Turner 2021: learned representations (neural activity) and functional properties (purpose). Specifically, we  extensively analyzed the representations used by the similarity-penalized RNNs, and how they differed from those of standard RNNs (e.g. dynamic versus static population codes for task variables, nonlinear encoding schemes, weight subspace and regression subspace overlap), and also extensively assessed their functional properties with task perturbations at test-time (Fig. 5).
>
> In terms of understanding the solutions at the level of dynamical properties, in line with previous studies, we focused on what is revealed by fixed point topology and measures of dynamical similarity. We acknowledge that we did not get a description of the alternative solutions' dynamics at the same level of granularity as is available for standard solutions to these tasks, but this is a direct consequence of the fact that a similarly reduced-order description is difficult or impossible for these more complex solutions. For instance, for context-dependent integration, standard solutions can essentially be described by the geometric relationships between a small set of vectors that do not appreciably change over time. This is afforded by the fact that for such solutions, the dynamics are pretty much always slow, and are confined to 1D manifolds, both of which we find to be not true for similarity-penalized solutions.
>
> ### Regarding similarity to experimental data:
> We would be shocked if there is a "yes" or "no" answer to this question, as the answer should depend on when the inductive biases of training for a particular task/function happen to align with biology, and when they do not. For the specific tasks we study, we already discuss the relation of the solutions we find to experimental data in the first two paragraphs of the Discussion. To recap briefly here:
>
> Standard RNNs ubiquitously find solutions to simple memory tasks that use persistent activity, yet experimental studies have found many cases of highly dynamic representations underlying memory functions (see line 61 for refs.). The dynamic, as opposed to static encoding of memories that we find for similarity-penalized RNNs appears to match this observation at a qualitative level.
>
> For context-dependent integration, the binary decision making variant of the task has been extensively studied experimentally in animal models. As noted in the Discussion, for this task, the alternative solutions we find and report here do appear to be less realistic than standard solutions in terms of their population coding properties. In particular, consistent with neural data, standard RNNs show very reliable linear decodability of the coherences of both task stimuli  regardless of selected context, yet this is not the case for similarity-penalized RNNs. However, we argue that also being able to find unrealistic solutions is crucial for probing when and why simple solutions align with biology, a point that we have now added to the Discussion (lines 467-468).

---

> > ### Author Response · Authors · 2025-11-20
> > **Response Part 2**
> >
> > ### Response to Questions:
> > * We thank the reviewer for bringing our attention to other studies that have considered related ideas. We have added the suggested references and have made note of these connections in lines 153-155: "This...activity."
> > * There is potentially a useful notion of "capacity", dependent on task complexity and network size, that could be defined in terms of how many iterations of INSD can be run before the RNN begins to either fail to satisfy the similarity penalty, or fails to solve the task well. For example, for the MemoryPro task, after two iterations of INSD, alternative solutions began to exhibit degraded task performance, yet for the other two tasks, task performance still remained high at this iteration. At the moment we can only speculate as to the minimal network size where multiple solutions that are not linearly predictive of each other can coexist, but agree that there is an interesting systematic study to be done there. We have expanded the final paragraph of the Discussion to reflect these thoughts: "More generally...across systems." (lines 496-500).
> > * While we considered standard rate-based RNNs for our experiments in this work, there is no reason why this approach can't be applied to models trained with intrinsic neuron-level noise or weight noise. We would expect that such modifications of the dynamics would further constrain the space of allowable solutions, and thus may suppress the aforementioned "capacity" of INSD. In terms of robustness of solutions once trained, the experiments in Fig. 5 demonstrate that, at least for task-relevant modifications of trial conditions (such as increased stimulus noise or lengthened trials), similarity-penalized solutions are generally on par with and sometimes moderately outperform standard solutions.

---

### Official Review · Reviewer_ReZK · 2025-11-01

**Soundness:** 3
**Presentation:** 3
**Contribution:** 3
**Rating:** 8
**Confidence:** 4

**Summary:**

The manuscript presents a new approach to train RNNs for the purpose of generating new hypotheses about neural mechanisms in neuroscience. Current approaches for training RNNs on neuroscience tasks tend to result in similar types of solutions, often relying on low-dimensional arrangements of stable fixed points and relatively simple dynamics around them. The authors’ use an iterative training procedure to find a family of RNNs that all solve a given task, but do so by implementing different dynamics and solutions. Specifically, each subsequent RNN in the family is trained with a loss that combines task accuracy with a penalty for solutions that are similar to those implemented by the previously trained networks. The authors apply this procedure to three well known and often used tasks of relevance to neuroscience and show that the resulting RNNs implement solutions that differ from those obtained with standard training approaches.

**Strengths:**

The paper is well written, clear, and technically strong.

Given the increasing use of RNNs as models of neural computations, the paper addresses an important question, and develops a creative approach that may increase the number of hypotheses that can be generated with RNNs.

The paper puts forward a new metric to compare the dynamics of different RNNs that overcomes some of the limitations of previously proposed metrics.

**Weaknesses:**

The dissimilarity measure employed by the authors to compare RNN dynamics overcomes some of the limitations of previously proposed measures, in particular by preventing RNNs to simply add irrelevant dynamics in new subspaces to otherwise unchanged dynamics analogous to that in the standard RNN solution. However, it is not clear to me whether their dissimilarity can prevent more subtle ways to superficially modify the dynamics without changing its essence.

For example, one could imagine a scenario where an alternative solution to the context-dependent integrator appears much more high-dimensional and “curved” than in the standard solution, and yet locally the nature of the interaction between the inputs and slow dynamics are preserved. If such a solution was possible, a comparison of the trajectories alone would not adequately capture the fact that locally the solutions are quite similar in essence.

More generally, whereas the standard solutions are well understood at a variety of levels of description (e.g. the local relative arrangement of inputs, left and right eigenvectors in the context dependent integrator), the authors do not reach the same level of understanding for the alternative solutions. Without such a full understanding, it is challenging to decide how “different” the various solutions really are.
It does not help that the alternative RNN produce dynamics that seems substantially more complex than that in standard RNNs, and so understanding the underlying mechanisms (which ultimately is required to generate new hypotheses about brain function) may be more challenging.

The authors’ approach for generating alternative solutions is quite heavy handed, in that it forces solutions with very different trajectories, whereas it is known that the different mechanisms can produce very similar trajectories (recent Pagan et al work). Presumably, these different solutions could not be found with the approach presented in the manuscript.

**Questions:**

Can the authors show in more detail how the alternative RNN solve their tasks? In particular, can they explain why the additional dynamics that distinguishes them from the standard networks is required to solve the task?

Standard RNNs can often be understood with fixed points and dynamics around them, and the authors argue that the same descriptions to not apply to the alternative RNNs. But then how can these be understood and used to generate testable hypotheses?

Can the authors’ approach be used to identify the various solutions to context dependent integrations found by Pagan et al?

---

> ### Author Response · Authors · 2025-11-20
> **Response Part 1**
>
> We thank the reviewer for their thoughtful comments, and are grateful for their positive assessment of our work. Since there is a lot of overlap between points raised under Weaknesses and Questions, we address them together below:
>
> First, before discussing dynamics, we believe we provided extensive evidence that the population coding properties with respect to task-relevant variables in similarity-penalized RNNs are decidedly different from those of standard RNNs (e.g. dynamic versus static population codes for task variables, nonlinear encoding schemes, weight subspace and regression subspace overlap, Fig. 2 and Fig. 4). Population coding is a property of solutions that is analogous to what experimentalists would most directly be able to observe and (in)validate, and thus we find it worth emphasizing that it is already useful to find solution diversity at this level.
>
> ### Concerns that alternative solutions could still be dynamically similar:
> We acknowledge that we did not get a description of the alternative solutions' dynamics at the same level of granularity as is available for standard solutions, but this is a direct consequence of the fact that a similarly reduced-order description is difficult or impossible for these more complex solutions. For instance, as you note for context-dependent integration, standard solutions can essentially be described by the geometric relationships between a small set of vectors that do not appreciably change over time. This is afforded by the fact that for such solutions, the dynamics are pretty much always slow, and are confined to 1D manifolds, both of which we find to be not true for similarity-penalized solutions.
>
> It is certainly possible that at some level---not discerned by the slow/fixed point analysis we did and the dynamical similarity analysis metric we used---that there is a way to relate the dynamical mechanisms used by similarity-penalized RNNs to those used by standard RNNs. That said, the performance profiles of standard and similarity-penalized RNNs on various out-of-distribution modifications of the tasks (shown in Fig. 5) precludes the possibility that the solutions are functionally equivalent, and thus precludes the possibility that the dynamics crucial for the task are actually equivalent in some hidden way.
>
> ### The heavy-handedness of our approach and our work in the context of Pagan et al.:
> This is a fair point, although we are not claiming that our approach is a universal way to uncover all solutions that might exist at every level of granularity of solution degeneracy. It does however jumpstart the process of escaping the simplicity bias. For context-dependent integration, we argue that the solutions we find are actually outside the triangle of solutions visualized in Pagan et al., as all solutions defined there assume the premise of two lines of fixed points that are integrated along. Although we did not do an extensive sweep over the strength of the similarity penalty, solutions that differ from standard solutions in more fine-grained ways may be discoverable if the similarity penalty is set to be small. Since we are penalizing something about neural activity, our approach likely cannot fully capture degeneracies that live at the level of the relative arrangements between left eigenvectors and input vectors, but may be able to promote degeneracy in the relative arrangements of the right eigenvectors (line attractor directions) across contexts, which do directly manifest in activity. We leave a more detailed investigation of this idea for future work, and have now commented on this limitation of our approach in the Discussion: "While we...activity." (lines 482-489).

---

> > ### Author Response · Authors · 2025-11-20
> > **Response Part 2**
> >
> > ### Deeper understanding of the dynamical properties of alternative solutions:
> > In terms of deeper understanding of how the alternative solutions use their unique dynamical properties to solve tasks, taking the MemoryPro task as an example, we observed that angles are stored during the memory phase in limit-cycle-esque trajectories confined to the output nullspace, with trajectories corresponding to different target angles precessing at slightly different speeds or phases. Admittedly, this is perhaps not a fully satisfactory description of the dynamical mechanism, as the precise manner of the interaction between input vectors and oscillatory eigenmodes to give rise to this encoding scheme was not unpacked in this paper. However, because the standard solution uses much simpler dynamics, one perhaps may have been satisfied there at the level of understanding of "inputs push the network state onto a ring of fixed points that encodes memories".
> >
> > We focus the present work on presenting a systematic method for finding alternative solutions, and providing extensive evidence that the solutions it finds are indeed different when sliced along multiple commonly used levels of understanding (encoding of task-relevant information, arrangements of slow/fixed points, and functional properties). An even more in-depth interpretation of the mechanisms used by the alternative solutions found is also important, but would require significantly more sophisticated analysis to be fully satisfactory due to their complexity; this is somewhat secondary to the goal of this paper. Nevertheless, we agree that describing these alternative solutions further and more precisely isolating how their unique dynamical properties contribute to solving tasks is a fruitful goal and something we will certainly pursue in future work.

---

### Meta-Review · Area_Chair_vsZB · 2026-01-09

**Summary:**

##### Reviewer ReZK

(1) The diversity of possible implementations with functional similarity is not captured by the method.

##### Reviewer SAv4

(1) The paper focuses on implementational diversity at the expense of understanding the functions the RNNs implement.

##### Reviewer 657y

(1) The settings are toy.

**Reviewer Concerns:**

The concerns of all reviewers appear to be addressed by the thorough author response.

**Reviewer Scores:**

Since the concerns of the reviewers were primarily about the conceptual focus of the work, I believe that some if not all would have updated their score or maintain their already high score. This paper is thus an easy accept.

---

### Decision · Program_Chairs · 2026-01-26

Accept (Poster)